# Design and evaluation of a mechanical pencil-based actuator for a wasp-inspired needle

Jette Bloemberg[1]*, Mario van der Wel[2], Aimée Sakes[1], Paul Breedveld[1]

**1** Bio-Inspired Technology (BITE) Group, Department of BioMechanical Engineering, Faculty of Mechanical Engineering, Delft University of Technology, Delft, The Netherlands, **2** Department of Electronic and Mechanical Support Division, Faculty of Electrical Engineering, Mathematics and Computer Science, Delft University of Technology, Delft, The Netherlands

* J.Bloemberg@tudelft.nl

## Abstract

In percutaneous interventions, long and thin needles are used to reach deep target locations within the body. However, inserting a long and thin needle into the tissue can cause needle buckling, resulting in poor control of the needle's trajectory and reduced targeting accuracy. In nature, the female parasitic wasp prevents the buckling of her long and slender ovipositor through a self-propelled motion. This study presents a stationary actuation system that can advance a wasp-inspired self-propelled needle consisting of seven 0.3-mm stainless steel rods with a theoretically unlimited insertion length. Based on the pencil lead advance mechanism in mechanical pencils that advances the pencil lead at a fixed increment when the pencil button is pushed, our actuation system advances the seven needle segments that comprise our needle by locking, advancing, releasing, and retracting the advance mechanisms. Experimental evaluation demonstrated that the actuation system successfully executes these actions, enabling step-by-step propulsion of the needle segments in gelatin-based tissue-mimicking phantoms. Moreover, the needle achieved mean motion efficiencies of 98±2%, 68±5%, and 57±7% in air, 5-wt% gelatin, and 10-wt% gelatin, respectively, over 15 actuation cycles. This actuation system prototype, which is based on a mechanical pencil, is a step forward in developing self-propelled needles for targeting deep tissue structures.

## 1. Introduction

Needles serve as passageways to target specific locations within the body, facilitating a wide range of applications, such as extracting tissue samples for biopsies and positioning instruments for needle-based therapies in deep tissue structures such as the prostate gland. When a needle is pushed through the tissue, forces arise at the needle tip and along the needle shaft [1]. Okamura *et al.* [1] demonstrated that inside homogeneous tissue, these forces comprise cutting and friction forces. To move a needle through tissue, the operator should apply a force that overcomes the

**Data availability statement:** All relevant data are within the manuscript and its Supporting Information files.

**Funding:** This work was supported by the Netherlands Organisation for Scientific Research (Nederlandse Organisatie voor Wetenschappelijk Onderzoek, NWO), domain Applied and Engineering Sciences (TTW), and which is partly funded by the Ministry of Economic Affairs. Project number P18-26 Project 4: Novel Trans-Perineal Laser Ablation (TPLA) Platform for accurate treatment of prostate tumours under MRI, Perspectief programme, Photonics Translational Research – Medical Photonics (MEDPHOT), was awarded to PB. URL: https://www.nwo.nl/. The funders had no role in study design, data collection and analysis, decision to publish, or preparation of the manuscript.

**Competing interests:** The authors have declared that no competing interests exist.

sum of these forces acting on the needle. If the axial force on the needle tip exceeds the critical load of the needle, the needle buckles [2]. Needle buckling might lead to poor control of the needle path, thereby decreasing needle targeting accuracy, which affects the effectiveness of biopsies and needle-based therapies [3].

To prevent needle buckling, self-propelled needles inspired by the ovipositor of the female parasitic wasp have been developed [4–11] (Fig 1a, Table 1). Wasp-inspired self-propulsion inside a substrate is accomplished by a set of parallel needle segments that can advance with respect to one another (Fig 1b). The advancing needle segments experience both a cutting force at their tips and a friction force along the length of their shafts in contact with the surrounding substrate [12]. The non-advancing needle segments, however, only experience a friction force in the opposite direction. If the sum of the friction and cutting forces on the advancing needle segments is equal to the sum of the friction forces on the non-advancing needle segments, the wasp-inspired needle can propel forward through a substrate with a zero net insertion force (Eq 1), as shown in previous works [4–11]. This can be achieved by keeping the number of advancing needle segments smaller than the number of non-advancing needle segments so that the difference between the forces acting on the two groups of needle segments increases with the insertion distance.

$$\sum_{i=1}^{a} \left( \mathbf{F}_{\text{fric,adv},i} + \mathbf{F}_{\text{cut,adv},i} \right) \leq \sum_{j=1}^{n} \left( \mathbf{F}_{\text{fric, non-adv},j} \right)$$

(1)

Where $a$ is the number of advancing needle segments, $n$ is the number of non-advancing needle segments, $\mathbf{F}_{\text{fric,adv}}$ is the friction force along the shafts of the advancing needle segments, $\mathbf{F}_{\text{cut,adv}}$ is the cutting force on the tips of the advancing needle segments, and $\mathbf{F}_{\text{fric, non-adv}}$ is the total amount of friction along the shafts of the non-advancing needle segments, which works in the opposite direction to the friction force of the advancing needle segments.

In existing prototypes of ovipositor-inspired needles, the needle and actuation system are integrated [4–11]. As a result, when the needle advances into the tissue, the actuation system must move in conjunction with it. The inertia of the moving components adversely affects the needle's self-propelled motion. To address this issue, Bloemberg et al. [14] developed an actuation system that keeps the external parts of the actuation system stationary, allowing only the needle and internal parts to move. However, a limitation of this mechanism is that the length of the actuation system restricts the achievable needle insertion distance. Therefore, this study aims to design a stationary actuation system capable of advancing a wasp-inspired self-propelled needle over very long insertion distances.

## 2. Design

### 2.1. Needle

Our design comprises a needle and a motorized actuation system. The needle consists of seven parallel needle segments inspired by the parallel valves of the parasitic wasp ovipositor (Fig 1b). To accommodate a central functional needle segment, such

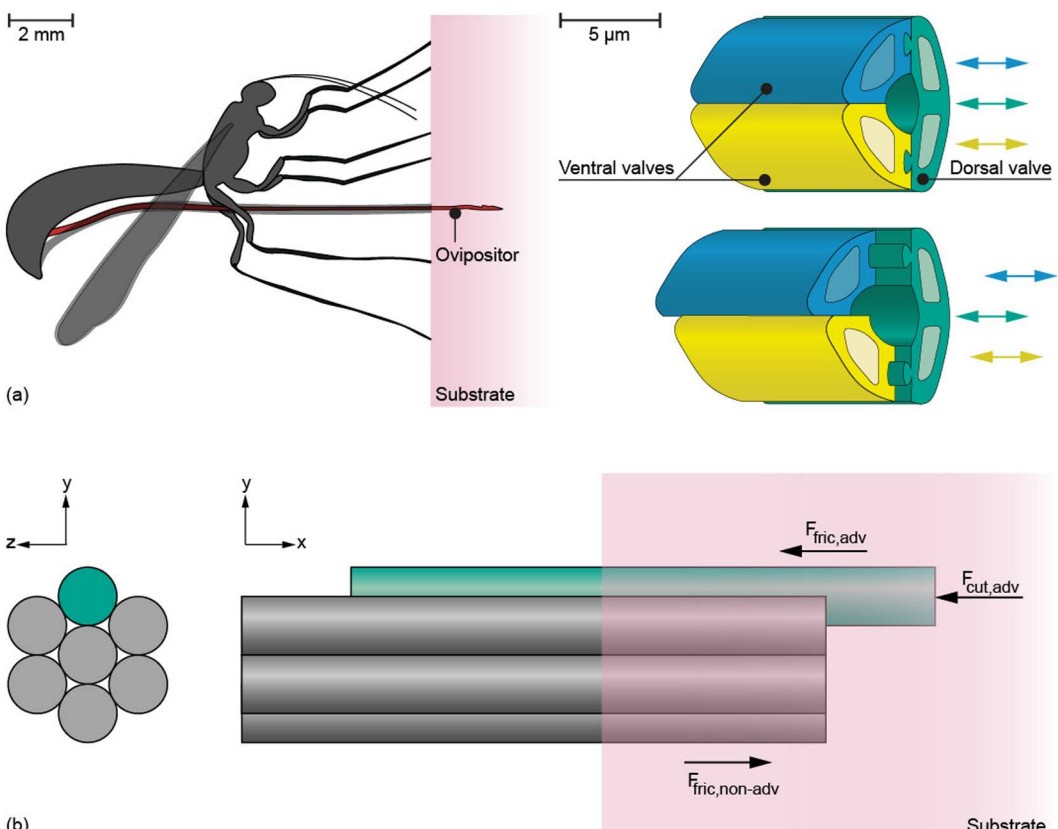

**Fig 1. Schematic representation of ovipositor-inspired needle motion. (a)** Schematic illustration of a female parasitic wasp using its ovipositor to lay eggs inside a substrate. The ovipositor consists of three parallel "valves" (green, yellow, and blue) that can move reciprocally (based on Cerkvenik *et al.* [13]). **(b)** Schematic illustration of ovipositor-inspired needle insertion into tissue with one advancing needle segment (green) and six non-advancing needle segments (gray). $F_{fric,adv}$ is the friction force along the advancing needle segment, $F_{cut,adv}$ is the cutting force on the tip of the advancing needle segment, and $F_{fric,non-adv}$ is the total amount of friction of the non-advancing needle segments, which works in the opposite direction to the friction force of the advancing needle segment.

**Table 1. State-of-the-art in wasp-inspired needles, showing the reference, the number of needle segments the needle comprises, the outer diameter of the needle in [mm], the material of the needle segments, and a description of the needle design.**

| Reference | Number of needle segments | Outer diameter [mm] | Material | Needle design |
|---|---|---|---|---|
| Frasson *et al.* [4] | 2 | 4.4 | Tango Black™ | Axially interlocked halves |
| Leibinger *et al.* [5] | 4 | 4 | VeroGrey | Axially interlocked quadrants |
| Scali *et al.* [6] | 7 | 1.55 | Nitinol | Converging/diverging ring with 7 holes interlocks the needle segments |
| Scali *et al.* [8] | 7 | 1.2 | Nitinol | Flower-shaped ring with 7 holes interlocks the needle segments |
| Scali *et al.* [7] | 6 | 0.84 | Nitinol | Heat shrink tube interlocks the needle segments |
| | 3 | 0.59 | | |
| | 6 | 0.42 | | |
| Bloemberg *et al.* [9,10] | 6 | 0.84 | Nitinol | Heat shrink tube interlocks the needle segments |
| Bloemberg *et al.* [14] | 6 | 0.84 | Steel | Heat shrink tube interlocks the needle segments |
| Bloemberg *et al.* [11] | 7 | 3 | Nitinol | Internal ring through needle segment slots interlocks the needle segments |

as an optical fiber, we opted for six outer needle segments surrounding a seventh segment, which can be substituted with the functional segment. In the two-dimensional cross-section, arranging six cylindrical needle segments concentrically around the seventh needle segment forms an optimal configuration when all needle segments have the same diameter. Each outer needle segment contacts the central needle segment as well as two adjacent needle segments. This minimizes the total cross-sectional area and results in the least empty space between the needle segments where tissue could potentially accumulate. Using the wasp's self-propelling principle, the needle self-propels through the substrate by incrementally advancing the seven needle segments forward one by one. In this study, the motion sequence that involves advancing all seven needle segments one time is referred to as one *actuation cycle*. The distance each needle segment travels per actuation cycle is called the *needle stroke distance*.

## 2.2. Actuation system

To achieve long needle insertion distances through incremental steps, the actuation system must sequentially advance the individual needle segments. For each needle segment, this can be accomplished via a clamp that operates in a four-step cycle: (1) locking, (2) advancing, (3) releasing, and (4) retracting (Fig 2). First, the clamp (in yellow) locks around the needle segment, thereby clamping the needle segment at a defined position (in red). Second, the locked clamp advances, thus advancing the needle segment forward (in dark green). Third, the clamp releases the needle segment at the new position (in orange). Finally, the clamp moves backward to its original position (in light green), whereas the needle segment remains in its propelled position. Once the clamp arrives at its original position, it can clamp the needle segment at its new propelled position (in red). This cycle repeats until the needle segment arrives at its destination.

The incremental needle segment advance mechanism mirrors the functionality of a mechanical pencil, where pressing the button advances the pencil lead by a fixed amount (Fig 3). The mechanical pencil comprises five complex-shaped components that can move with respect to each other: (1) a housing, lead retainer, and nut portion (in gray and pink), (2) a button, central tube, and rear tube (in dark and light green) formed around a three-finger collet (in yellow), (3) a sleeve (in orange), (4) a compression spring (in blue), and (5) a pencil lead (in black). Among these components, the advance mechanism (in lighter colors) consists of the nut portion, the rear tube, the collet, the sleeve, and the spring. The nut portion

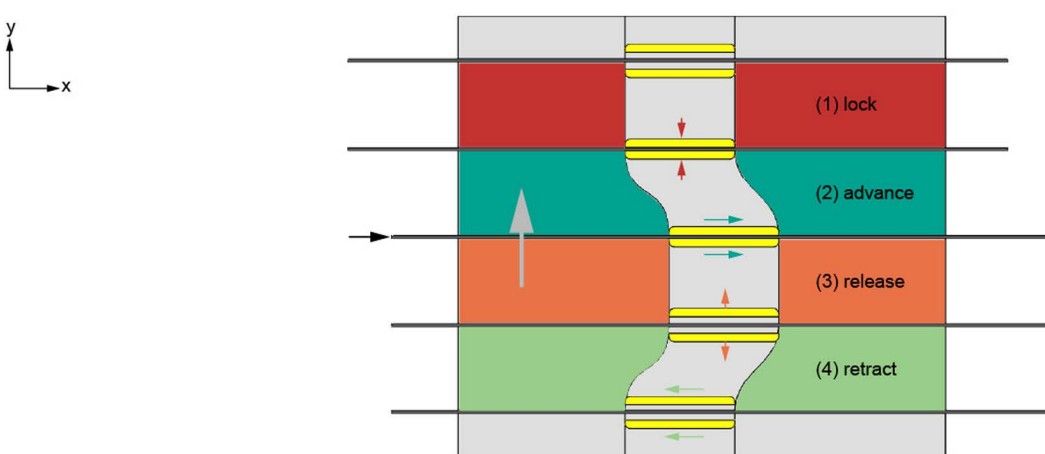

**Fig 2. 2D schematic representation of the clamp mechanism to achieve needle actuation in incremental steps.** The figure shows the simplified motion sequence of clamps actuated by a cam track (gray). During the red phase of the cam, the clamp (yellow) is locked around the needle segment (black). During the dark green phase of the cam, the clamp follows the cam track in the positive x-direction, thereby moving the needle segment in the positive x-direction. During the orange phase of the cam, the needle segment is released. During the light green phase of the cam, the clamp follows the track in the negative x-direction while the needle segment remains in its propelled position.

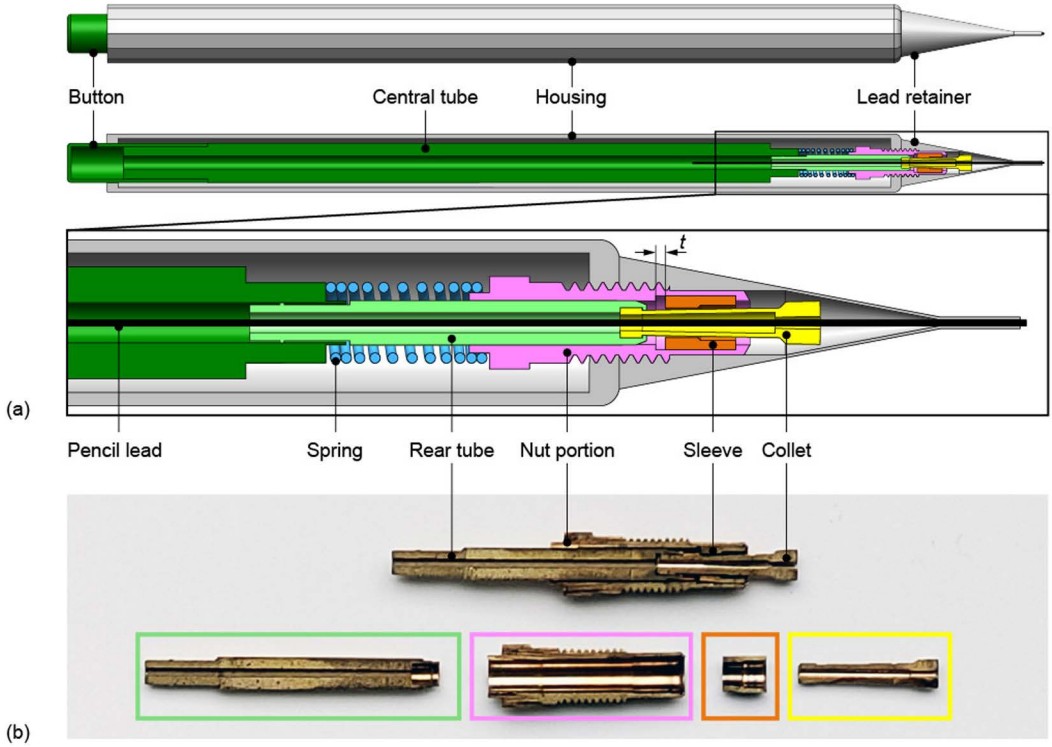

**Fig 3. The pencil lead advance mechanism of a mechanical pencil.** (a) Schematic illustration of the assembly and cross-section view of the pencil lead advance mechanism in its released state showing the button and central tube (dark green), housing and lead retainer (gray), pencil lead (black), spring (blue), rear tube (light green), nut portion (pink), sleeve (orange), and collet (yellow). Distance $t$ indicates the distance over which the sleeve can move within the nut portion, constrained by the ridges of the nut portion. (b) Picture of the cross-section of a pencil lead advance mechanism highlighting the rear tube (green), nut portion (pink), sleeve (in orange), and collet (yellow).

contains two ridges that limit the horizontal translation of the sleeve over a distance $t$ (Fig 3a). The sleeve is positioned around the collet. The collet is similar to that found in milling machines and closes around the pencil lead (in black). When the sleeve is positioned at the tip of the collet, the collet's fingers are closed, and the pencil lead is held firmly in place for writing. Conversely, when the sleeve is retracted from the tip of the collet, as shown in Fig 3a, the fingers of the collet are opened, and their grip on the pencil lead is released, allowing it to move.

To explain the mechanism of the mechanical pencil, we assume that the first component (i.e., the housing, lead retainer, and nut portion) remains stationary while the button is pressed. Pressing the button on the mechanical pencil causes the advance mechanism to move from its locked to its released state by translating the second component (i.e., the button, central tube, rear tube, and collet), causing the spring to compress. The sleeve also moves with the collet over distance $t$ until the sleeve encounters the right ridge of the nut portion. Until this moment, the sleeve was positioned at the tip of the collet because of the friction between the sleeve and the collet, thereby closing the collet's fingers and advancing the pencil lead firmly held by the collet. The right ridge of the nut portion limits the sleeve's forward motion, determining the pencil lead's advancement distance. When the sleeve is halted, the rear tube and the collet move further forward, overcoming the friction between the sleeve and the collet and pushing the collet's fingers out of the sleeve, causing the collet's fingers to open and release the pencil lead. The lead retainer at the tip of the pencil prevents the pencil lead from falling out of the pencil or retracting due to the frictional force between the lead retainer and the pencil lead.

Once the button is released, the spring extends, and the advance mechanism moves from its released state to its locked state by retracting the central tube, rear tube, and collet. The collet's retraction pulls the sleeve back over distance *t* into its original position within the nut portion. The collet's fingers remain open until the sleeve encounters the left ridge of the nut portion, closing the collet's fingers again around the pencil lead. The pencil lead remains extended despite the collet's movement due to greater friction between the pencil lead and the lead retainer than between the pencil lead and the open collet. Without the lead retainer, the frictional force between the pencil lead and the collet would cause the pencil lead to retract with the collet during its backward motion. Finally, the spring force causes the collet to retract even further to its original position, and the sleeve closes the collet's fingers around the pencil lead at its propelled position, securing the pencil lead and preventing it from retracting during writing.

We adapted an existing pencil lead advance mechanism of a mechanical pencil as the base for our needle segment advance mechanism. Rather than advancing a pencil lead, our needle segment advance mechanism enables the movement of a needle segment with the same diameter as the pencil lead. The advance mechanism is in its locked state when the spring is extended, thereby clamping the needle segment. Conversely, when the spring is compressed, the advance mechanism is in its released state, allowing it to translate in the longitudinal direction to clamp the needle segment at its new position.

In our actuation system design, seven advance mechanisms, arranged in a circle, drive the seven needle segments. To actuate the advance mechanisms, we use a cam containing two tracks that are followed by sliders attached to the distal and proximal ends of the advance mechanisms. The spring within each advance mechanism ensures continuous contact between the sliders and the cam tracks. A frame encasing the cam constrains the movement of the sliders to solely a translation in the horizontal x-direction. Consequently, the motion of the advance mechanisms is restricted to solely a translation in the x-direction, guided by the cam tracks.

To explain the motion cycle of our actuation system, the mechanism is simplified and visualized in a schematic illustration in Fig 4. The input motion is a rotation of the cam (in light gray) around its longitudinal x-axis. The cam controls the position of the proximal and distal sliders (in dark gray) that operate the advance mechanisms. The cam tracks govern the motion of each advance mechanism through four distinct phases of rotation: the red, dark green, orange, and light green phases, similar to Fig 2. First, the proximal slider of the advance mechanism moves in the negative x-direction, causing the spring to extend and the advance mechanism to lock the needle segment (red phase). We added the second and fourth cam phases because the pencil lead's advancement distance *t* is too short for our needle stroke distance. In the second cam phase, both the proximal and distal sliders move in the positive x-direction, moving the advance mechanism in its locked state forward and thus advancing the locked needle segment (dark green phase). Third, the distal slider remains stationary as the proximal slider continues to move in the positive x-direction, compressing the spring and releasing the needle segment (orange phase). In the fourth and final phase, both the proximal and distal sliders move in the negative x-direction, retracting the advance mechanism in its released state while the released needle segment remains in its propelled position (light green phase). Once the advance mechanism returns to its original position, it locks the needle segment at its new propelled position (red phase). This cycle repeats until the needle segment reaches its target location.

## 2.3. Final design

The final design contains seven advance mechanisms, one for each needle segment, and each is operated by two sliders that follow the cam tracks (Fig 5). To facilitate the self-propelled motion, only one of the seven needle segments is in the dark green phase during each step of the actuation cycle, ensuring that the number of advancing needle segments remains less than the number of non-advancing segments. The cam's dark and light green phases each provide a 4-mm stroke. The orange and red phases result in a 3-mm stroke for releasing and locking the advance mechanisms. The corners of the cam tracks between the phases were rounded to facilitate smooth transitions between the cam phases. One

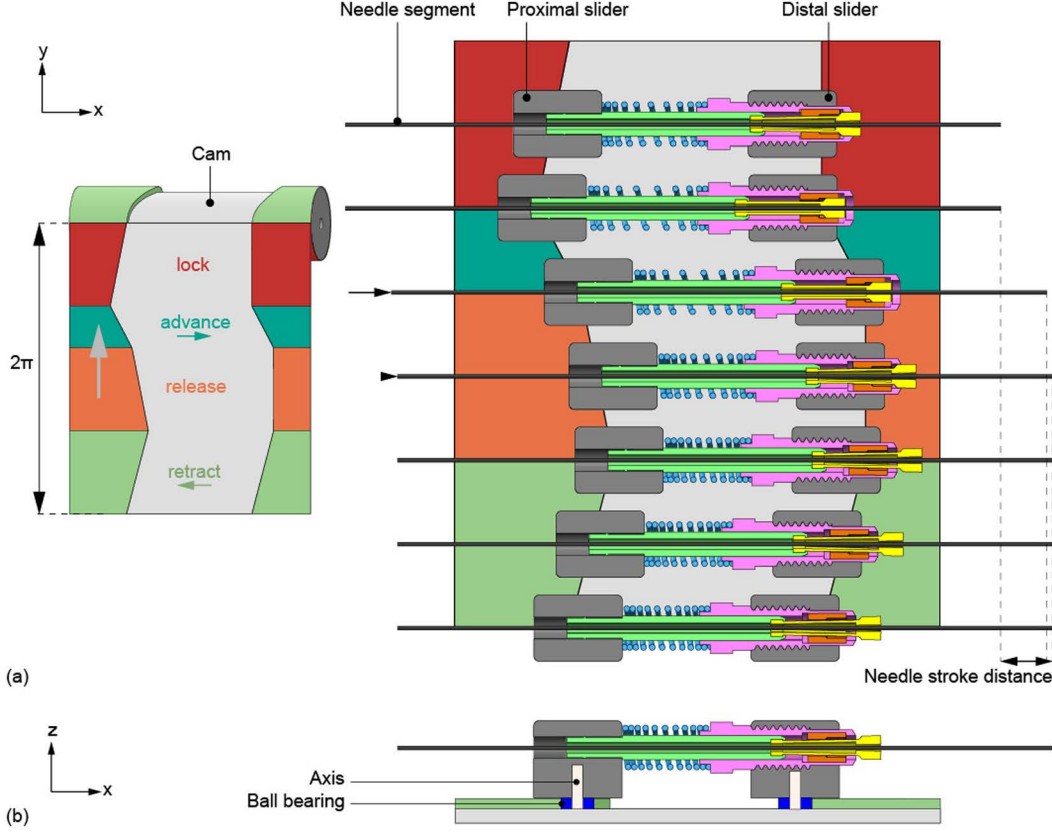

**Fig 4. Schematic representation of the needle segment advance mechanism actuated by a cam. (a)** The rolled-out cam shows the needle segment advance mechanism that comprises seven adapted pencil lead advance mechanisms (shown in the cross-section top view) that function as the clamp mechanisms actuated by the cam. **(b)** Cross-section side view of one needle segment advance mechanism with the proximal and distal sliders (dark gray) running in the cam tracks using ball bearings (dark blue) on axes (white).

actuation cycle is defined as a single full rotation of the cam, during which all seven needle segments are advanced once over the needle stroke distance.

In the actuation system, the needle segments run at a larger diameter than at the needle tip. The seven needle segments are arranged in a circular formation within the actuation system, whereas at the needle tip, one segment is centrally positioned with six segments arranged in a circle around it. To guide the seven needle segments smoothly from the actuation system to the needle tip, seven S-shaped tubes followed by seven straight tubes were used. The needle segments run through the S-shaped tubes that gradually decrease the distance between the needle segments from the actuation system to the needle tip. These tubes provide continuous support to the needle segments to avoid buckling while allowing free movement along the x-axis. The seven straight tubes further guide the needle segments into their insertion formation, where one segment is centrally located and six segments surround it. The friction between the tubes and the needle segments exceeds the friction between the open collets and the needle segments during the light green phase as the advance mechanisms retract. Consequently, the needle segments stay at their propelled positions while the collets are retracted.

## 2.4. Prototype

The actuation system design limits the needle segment diameter to 0.3 mm but does not impose any restrictions on the needle segment length. The needle used in this study consists of seven blunt spring steel rods with a diameter of 0.3 mm

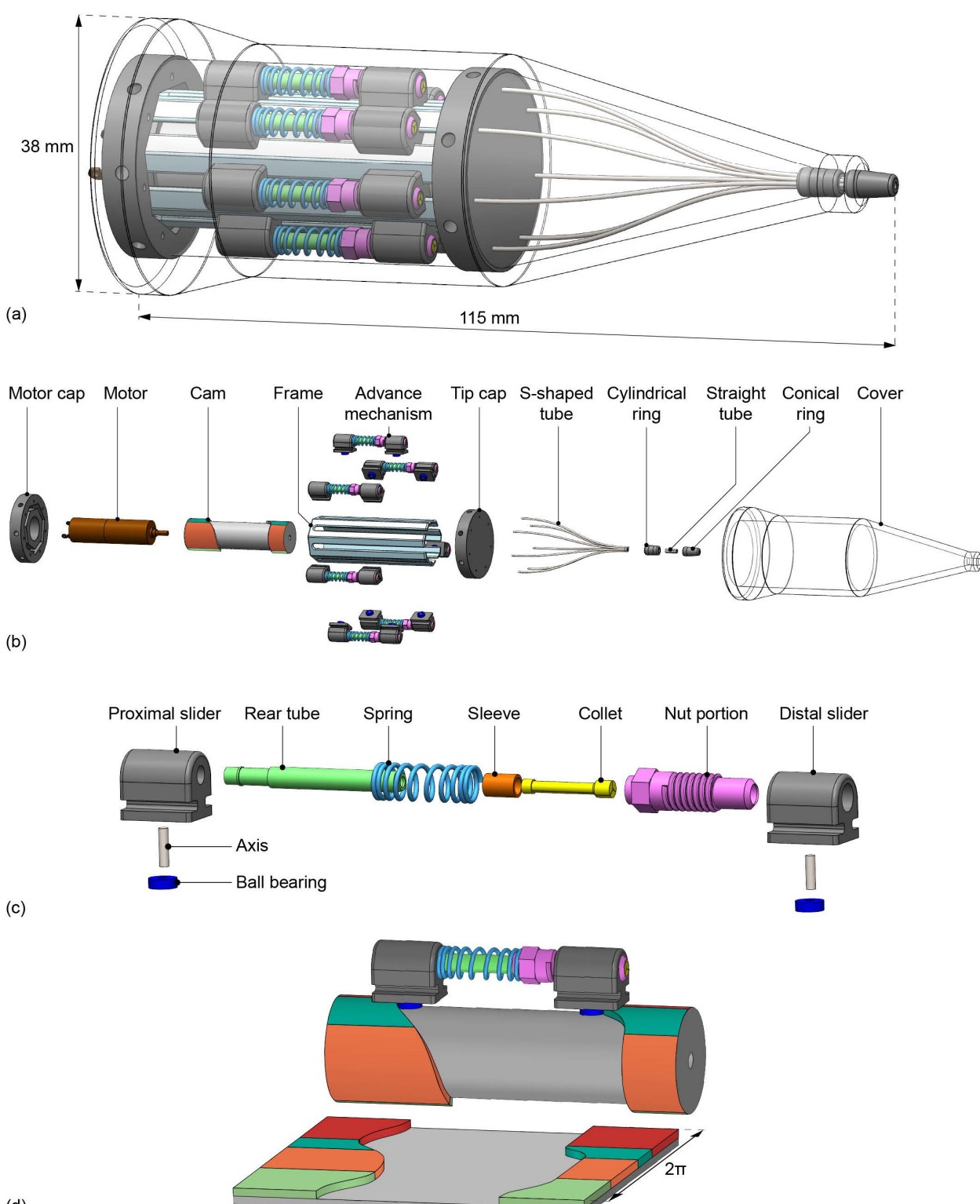

**Fig 5. Design drawing of the final design. (a)** Assembly. **(b)** Exploded view. **(c)** Exploded view of the advance mechanism adapted from the Pentel P203 pencil lead advance mechanism. **(d)** Cam showing one of the seven advance mechanisms in the cam tracks and rolled-out cam.

and a length of 1 m. Following the designs of Scali *et al.* [7] and Bloemberg *et al.* [9,10,14], the needle segments were held together at the tip using a 10-mm long heat shrink tube (*103–0352*, Nordson Medical Corp., Westlake, OH, United States). This bundling mechanism prevents the needle segments from diverging while only slightly increasing the overall diameter of the needle. To maintain its position at the needle tip, the heat shrink tube was secured to one of the needle segments using *Pattex Gold Gel* (*1432562*, Pattex, Henkel AG and Co., Düsseldorf, Germany). The remaining needle segments can move freely through the heat shrink tube. The total diameter of the needle, including the heat shrink tube, is 1.0 mm.

The assembled prototype, called the Ovipositor Needle Clamp Actuator (ONCA), is shown in Fig 6. Components of the needle segment advance mechanisms were adapted from off-the-shelf *Pentel P203 mechanical pencils* (Pentel Co. Ltd.,

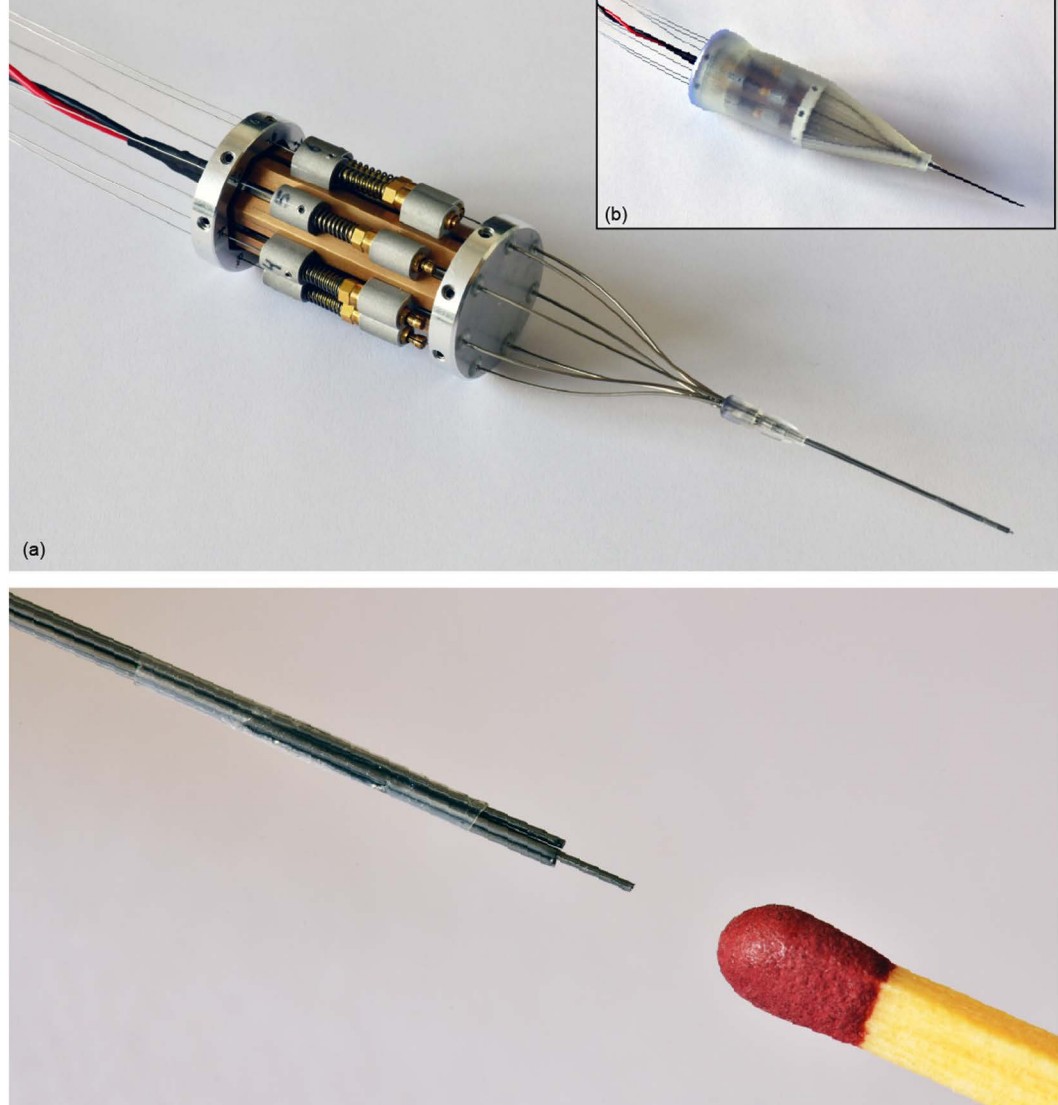

**Fig 6. Ovipositor Needle Clamp Actuator (ONCA). (a)** Without cover. **(b)** With cover. **(c)** Close-up of the needle tip consisting of seven rods held together by a shrinking tube (Nordson Medical) glued to one of the seven rods.

Penteru Kabushiki Gaisha, Japan), specifically the rear tube, nut portion, sleeve, and collet. The rather stiff pencil springs were replaced with softer *DR970 springs* (Alcomex springs, Opmeer, Netherlands; $\varnothing_{outer} = 3.6$ mm, $\varnothing_{wire} = 0.4$ mm, $L_0 = 12.80$ mm, $K = 0.78$ N/mm, $L_c = 5.50$ mm) because the tactile feedback from the stiffer springs is unnecessary for our prototype. The proximal and distal sliders were produced out of Aluminum 7075-T6 by wire Electrical Discharge Machining (EDM), after which the holes for the needle segments and axes were drilled, and screw threads were added for fixation of the nut portions. The bearing axes were machined at length out of high-speed steel by wire EDM. The bearings are stainless steel *Deep Groove Ball Bearings DDL-310HA1P25LO1* (MinebeaMitsumi Inc., Tokyo, Japan).

The cam, motor cap, and tip cap were produced by CNC milling and turning of Aluminum 7075-T6. The frame was constructed from brass, providing low friction with the aluminum sliders. The outer heptagon shape of the frame and its slots for the sliders were CNC milled. Subsequently, the inner heptagon shape was machined by wire EDM. Finally, the frame was machined to its final length by wire EDM. The S-shaped tubes (inner diameter of 0.4 mm and outer diameter of 0.6 mm) were manually bent from stainless steel. The cover, cylindrical ring, and conical ring were produced by stereolithography 3D printing on a Formlabs Form 3B printer using Clear resin (Formlabs, Somerville, MA, USA). The motor used is a *1016M012G DC-micromotor* with a *10/1 256:1 gearbox* (Faulhaber, Schönaich, Germany).

## 3. Evaluation

### 3.1. Experimental goal

To evaluate the performance of the ONCA, we conducted experiments in air and in tissue-mimicking phantoms. The goal was to investigate the ONCA's insertion performance in terms of its motion efficiency, $\eta_m(C)$, over the number of actuation cycles, $C$. More specifically, we investigated how $\eta_m(C)$ was influenced by two factors: (1) the slip between the needle segments and the advance mechanisms, which reflects the actuation system's clamp efficiency, $\eta_c(C)$, and (2) the slip between the stationary needle segments and the substrate (i.e., air or tissue-mimicking phantoms), which reflects the needle's propulsion efficiency, $\eta_p(C)$.

- $C$ [-]: number of actuation cycles

- $\eta_m(C)$ [%]: motion efficiency of the ONCA

- $\eta_c(C)$ [%]: clamp efficiency of the advance mechanisms inside the actuation system of the ONCA

- $\eta_p(C)$ [%]: propulsion efficiency of the needle of the ONCA

For $\eta_c(C) = 100\%$, the advance mechanism clamps function perfectly, and the needle segments do not slip with respect to the advance mechanisms. We assumed that when the needle travels in the air, the advance mechanisms operate at $\eta_c(C) = 100\%$ and advance the needle segments over the needle stroke distance during every actuation cycle. When the needle travels in a tissue-mimicking phantom, the cutting and friction forces acting on the needle segments can decrease $\eta_c(C)$, thereby advancing the needle segments over less than the needle stroke distance during every actuation cycle.

For $\eta_p(C) = 100\%$, the needle advances into the substrate over the same distance that the advance mechanisms move the needle segments during each actuation cycle. This means that the advancing needle segments advance in the substrate while the non-advancing needle segments remain stationary with respect to the substrate, meaning there is no slippage between the stationary needle segments and the substrate.

### 3.2. Experimental facility

The experimental setup consists of the ONCA kept stationary in a holder and the substrate (i.e., a tissue-mimicking phantom) on a Perspex cart ([Fig 7a, b]). The cart fits on an air track (Eurofysica), facilitating near-frictionless linear translation of the cart. Millimeter paper was attached at the bottom of the cart and used as a reference for the traveled distance, $d_n(C)$, of the needle tip in the substrate on the cart. The variable $d_n(C)$ was measured by calculating the difference between

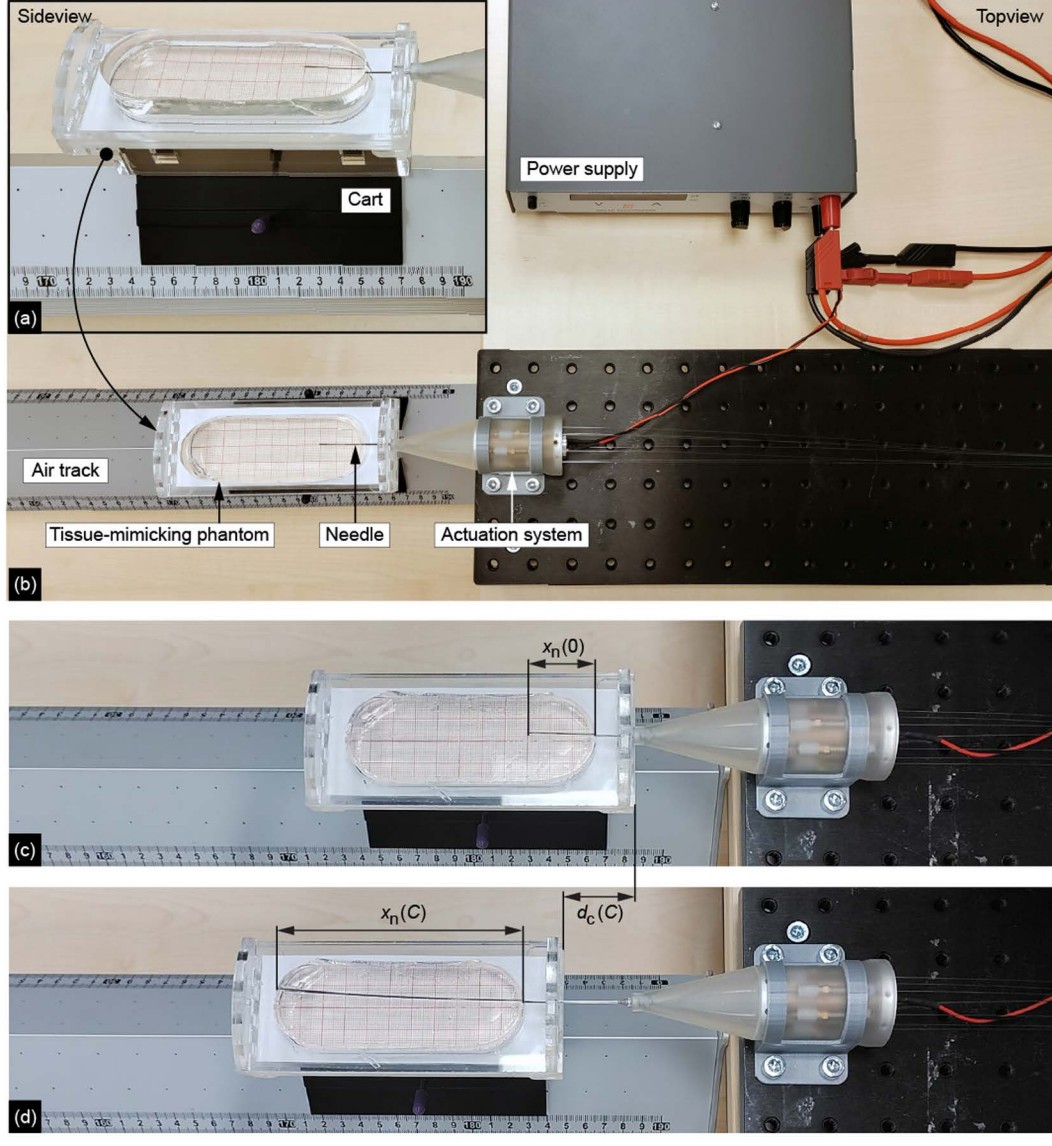

**Fig 7. Experimental setup. (a)** Side view of the cart in the experimental setup. **(b)** Top view of the experimental setup. **(c)** Example of the initial position of the needle inside the tissue-mimicking phantom. Variable $x_n(0)$ is the measured initial insertion distance of the needle in the substrate at zero actuation cycles. **(d)** Example of a final position of the needle inside the gelatin phantom. Variable $x_n(C)$ is the measured distance of the needle in the substrate after $C$ number of actuation cycles and $d_c(C)$ is the measured displacement of the cart with respect to the air track.

the initial needle insertion distance, $x_n(0)$, and the insertion distance, $x_n(C)$, in the substrate after a number of actuation cycles, $C$, from the video footage (Fig 7c, d). Moreover, we measured the distance $d_c(C)$ over which the cart was pushed forward with respect to the air track during needle insertion. The variable $d_c(C)$ was determined by calculating the difference between the initial cart position and the cart position after the number of actuation cycles, $C$, with respect to the air track from the video footage using the ruler on the air track (Figs 7c, d).

- $x_n(0)$ [mm]: initial needle insertion distance

- $x_n(C)$ [mm]: needle insertion distance

- $d_n(C)$ [mm]: traveled distance of the needle tip in the substrate

- $d_c(C)$ [mm]: traveled distance of the cart with respect to the air track

The distance over which the advance mechanisms advance the needle segments equals $d_n(C) + d_c(C)$. The expected traveled distance, $d_e(C)$, of the needle tip was computed by averaging the distance traveled by the needle tip with respect to the air track per actuation cycle in the air (i.e., the needle stroke distance) and multiplying it by $C$.

- $d_e(C)$ [mm]: expected traveled distance of the needle tip

### 3.3. Experimental variables

In the experiments, the independent variables were $C$, which was set to 5, 10, 15, and 20, and the substrate through which the needle was propelled, which was either air or tissue-mimicking phantoms. The concentration of gelatin powder in the tissue-mimicking phantoms was set at weight ratios (wt) of 5 wt% or 10 wt%. These weight ratios lead to gelatin samples with moduli of elasticity of approximately 5.3 and 17 kPa, respectively [7], resembling healthy liver tissue (<6 kPa) [15] and healthy muscle tissue (12–32 kPa) [16] or healthy prostate tissue (16 kPa) [17].

The dependent variables were $d_n(C)$ and $d_c(C)$. The dependent variables and $d_e(C)$ were used to compute $\eta_c(C)$ (Eq 2), $\eta_p(C)$ (Eq 3), and $\eta_m(C)$ (Eq 4).

$$\eta_c(C) = \frac{d_n(C) + d_c(C)}{d_e(C)} \cdot 100\%$$

(2)

$$\eta_p(C) = \frac{d_n(C)}{d_n(C) + d_c(C)} \cdot 100\%$$

(3)

$$\eta_m(C) = \eta_c(C) \cdot \eta_p(C) = \frac{d_n(C)}{d_e(C)} \cdot 100\%$$

(4)

### 3.4. Experimental protocol

Before each experiment run, the cart was manually translated over 30 mm to ensure $x_n(0) \approx 30$ mm, establishing sufficient contact between the needle surface and the substrate for the self-propelled motion. The initial manual insertion is required to enable the needle's self-propelled motion. After the initial manual insertion, the actuation system was turned on, allowing the needle segments to move within the substrate. The actuation system was turned off when the needle segments reached an insertion distance, $x_n(C)$, of 110 mm or when the actuation system ran for 60 seconds. Each experimental condition was repeated six times.

### 3.5. Results

The means and standard deviations of $d_n(C)$ and $d_c(C)$, as well as those of $\eta_c(C)$, $\eta_p(C)$, and $\eta_m(C)$, for each experimental condition are presented in Table 2 and Fig 8. In the air, the needle had already reached $x_n(C) = 110$ mm after an average of 18 actuation cycles. Hence, the traveled distances and efficiencies for more than 15 actuation cycles are provided only for the tissue-mimicking phantoms, as indicated by the gray overlay in Fig 8.

The data in Table 2 show that the mean $\eta_c(C)$ in 5-wt% gelatin remained approximately constant and comparable to the mean $\eta_c(C)$ in air over the 20 actuation cycles. In contrast, the mean $\eta_c(C)$ in 10-wt% gelatin decreased over the number of actuation cycles from 100±6% over the first five actuation cycles to 66±9% over 20 actuation cycles.

**Table 2. Results of the experiment, showing the substrate through which the needle traveled, number of actuation cycles, traveled distance of the needle tip in the substrate, traveled distance of the cart, clamp efficiency, propulsion efficiency, and motion efficiency, with mean values and standard deviations.**

| Substrate | Number of actuation cycles, ($C$) | Traveled distance needle tip [mm], $d_n(C)$ (mean±std) | Traveled distance cart [mm], $d_c(C)$ (mean±std) | Clamp efficiency [%], $\eta_c(C)$ (mean±std) | Propulsion efficiency [%], $\eta_p(C)$ (mean±std) | Motion efficiency [%], $\eta_m(C)$ (mean±std) |
|---|---|---|---|---|---|---|
| Air | 5 | 22±2 | 2±1 | 100±6 | 93±5 | 93±8 |
| | 10 | 44±2 | 2±1 | 100±3 | 96±2 | 96±4 |
| | 15 | 67±2 | 2±1 | 100±1 | 98±2 | 98±2 |
| 5-wt% gelatin | 5 | 13±2 | 10±2 | 95±2 | 56±8 | 53±7 |
| | 10 | 29±3 | 16±3 | 98±0 | 64±7 | 62±6 |
| | 15 | 47±3 | 21±4 | 98±1 | 70±5 | 68±5 |
| | 20 | 66±3 | 24±4 | 98±1 | 74±5 | 72±3 |
| 10-wt% gelatin | 5 | 18±1 | 6±1 | 100±6 | 77±3 | 77±5 |
| | 10 | 32±3 | 9±1 | 89±8 | 77±2 | 68±6 |
| | 15 | 39±5 | 13±2 | 76±9 | 75±2 | 57±7 |
| | 20 | 45±7 | 16±2 | 66±9 | 73±2 | 49±7 |

The mean $\eta_p(C)$ in air and 5-wt% gelatin increased with the number of actuation cycles, increasing from 93±5% and 56±8%, respectively, during the first five cycles, to 98±2% and 70±5% over 15 actuation cycles. In contrast, the mean $\eta_p(C)$ in 10-wt% gelatin remained approximately constant in the 73–77% range.

From the mean $\eta_m(C)$, we can see that in air and in 5-wt% gelatin, the mean percentages increased over the number of actuation cycles from respectively 93±8% and 53±7% over the first five actuation cycles to 98±2% and 68±5% over 15 actuation cycles. However, in 10-wt% gelatin, the mean $\eta_m(C)$ decreased from 77±5% over the first five actuation cycles to 57±7% over 15 actuation cycles.

## 4. Discussion

### 4.1. Main findings

In this study, we presented the design of a stationary actuation system that can advance a wasp-inspired self-propelled needle by clamping and advancing the parallel needle segments, resulting in a theoretically unlimited insertion length. However, achieving theoretically unlimited insertion length requires at least three parallel needle segments and a corresponding driving mechanism. Based on the pencil lead advance mechanism in a mechanical pencil that advances the pencil lead over a fixed increment when the pencil button is pressed, the ONCA advances the seven needle segments that comprise our needle one by one. The actuation system enables the needle to propel itself through stationary tissue-mimicking phantoms by locking, advancing, releasing, and retracting the needle segment advance mechanisms while the actuation system maintains its stationary position.

In an ideal scenario, $\eta_p(C) = 100\%$, meaning that the needle will only self-propel and not push the substrate forward ($d_c(C) = 0$ mm). When the advance mechanisms, in that case, operate at $\eta_c(C) = 100\%$, this leads to $\eta_m(C) = 100\%$. The results of this study suggest that for the needle operating in air, the advance mechanisms functioned at $\eta_c(C) = 100\%$ and $\eta_p(C)$ was nearly 100%. More specifically, the mean $\eta_p(C)$ in air was not 100%, because of a slight movement of the cart, which was probably caused by the friction between the moving needle segments and the cart.

When $\eta_p(C) < 100\%$, while $\eta_c(C) = 100\%$, $d_n(C)$ is less than $d_e(C)$, as the cart is pushed forward and also travels a certain distance ($d_c(C) > 0$ mm). This situation is shown in the results for the needle in 5-wt% gelatin, where the mean $d_n(C)$ was lower than that in air, although, in both 5-wt% gelatin and in air, the mean $\eta_c(C)$ percentages were nearly 100%. This can be attributed to slipping of the stationary needle segments in the substrate. Nevertheless, over the number of

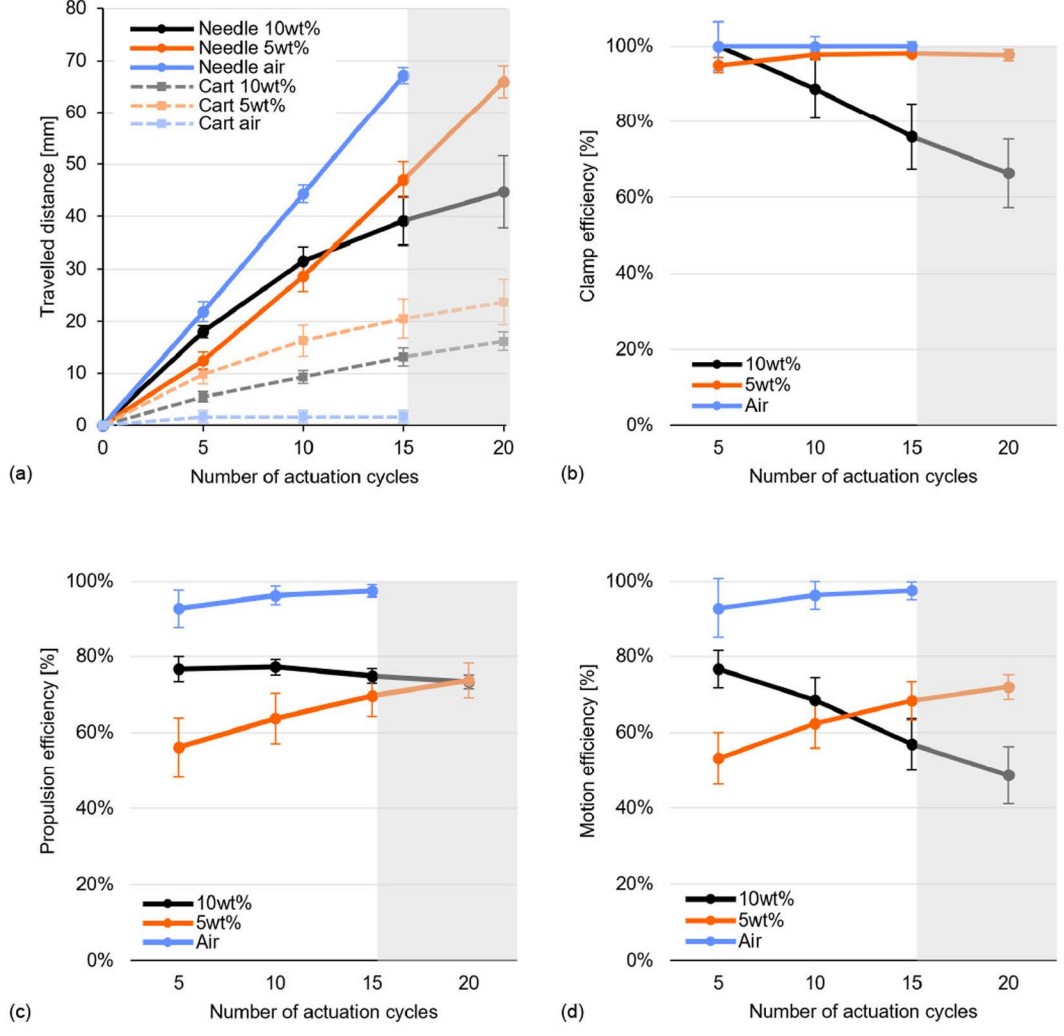

**Fig 8. Experimental results of (a) the traveled distance for the needle tip in the substrate ($d_n$) and the traveled distance of the cart ($d_c$) [mm], (b) clamp efficiency ($\eta_c$) [%], (c) propulsion efficiency ($\eta_p$) [%], and (d) motion efficiency ($\eta_m$)[%] over the number of actuation cycles ($C$).** The circles and solid lines are the mean values for the needle, the squares and dashed lines are the mean values for the cart, and the error bars are the standard deviation. The gray overlay indicates that the traveled distances and efficiencies for over 20 actuation cycles are provided only for the 5-wt% and 10-wt% gelatin-based tissue-mimicking phantom.

actuation cycles, the mean $\eta_p(C)$ in 5-wt% gelatin increased; thus, the slip in the substrate decreased. This is likely due to the friction force on the needle segments, which increased with increasing insertion depth, whereas the cutting force remained constant [18].

When the advance mechanisms operate at $\eta_c(C) <$ 100%, they fail to move the needle segments forward by $d_e(C)$ due to slipping within the clamps, which likely occurs during the dark green advance phase (i.e., when the collet is closed around the needle segment). This may be due to the cutting and frictional forces acting on the tip and the shaft of the advancing needle segment, which may be greater than the clamping force of the advance mechanism. This effect is shown in the experimental results for the needle in 10-wt% gelatin, where the measured mean $\eta_c(C)$ decreased with increasing number of actuation cycles, likely due to the increasing friction force on the needle segment with increasing insertion distance, which in turn increased the required clamping force.

To compare the performance of our needle with that of state-of-the-art wasp-inspired needles, we used $d_n(C)$ and $d_c(C)$ to compute the slip ratio ($s_{ratio}(C)$) of our needle in the substrate (Eq 5). $s_{ratio}(C)$ quantifies the slip between the stationary needle segments and the substrate, which limits the self-propulsion capabilities of the needle.

$$s_{ratio}(C) = 1 - \frac{d_n(C)}{d_n(C) + d_c(C)}$$

(5)

The mean $s_{ratio}(20)$ values of our needle of $0.26 \pm 0.045$ and $0.27 \pm 0.019$ in 5-wt% and 10-wt% gelatin, respectively, are greater than the slip ratios reported in previous research on wasp-inspired needles of $0.18 \pm 0.011$ [14] and 0.2 [7] in 5-wt% gelatin, and $0.19 \pm 0.007$ [14] and 0.3 [7] in 10-wt% gelatin. The differences in the slip ratios can be attributed to several factors. First, an actuation system using low-friction ball splines applies a small insertion push force onto the needle segments, which helps the advancing needle segment overcome the cutting and friction forces in the stationary tissue-mimicking phantom [14]. Consequently, this approach likely resulted in a lower slip ratio than our approach did. Second, Scali *et al.* [7] used fewer needle segments (six compared with seven) with smaller diameters (0.25 mm compared with 0.3 mm) than we did, leading to an overall smaller needle diameter than our needle (0.8 mm compared with 1.0 mm). A smaller needle diameter reduces forces at the tip of the needle, resulting in a lower slip ratio. Previous research has confirmed that the peak axial needle insertion force increases with increasing needle size [19].

### 4.2. Limitations

Among the seven parallel needle segments comprising our needle, the segment with the heat shrink tube (i.e., the bundling segment) exhibited a shorter traveled distance than the other segments did, resulting in a lower measured mean $\eta_c(C)$ for the bundling segment. The resultant $\eta_c(C)$ of the bundling segment (mean $\pm$ std) over the total insertion distance was $59 \pm 13\%$ and $15 \pm 2\%$ in the 5-wt% and 10-wt% gelatin samples, respectively. The shorter traveled distance of the bundling segment is likely caused by the increased friction and cutting forces compared with those of the other segments. These forces may overcome the clamping force of the needle segment advance mechanism, leading to the bundling segment slipping within the advance mechanism and causing a decrease in $\eta_c(C)$. As the bundling segment lags behind the other needle segments, the other needle segments begin to protrude while they are no longer bundled at the tip. This protrusion allows the needle segments to diverge at their tips, which increases the force required to advance the bundling segment over the other needle segments. Consequently, the lag of the bundling segment worsens with increasing insertion distance, decreasing $\eta_c(C)$ with increasing number of actuation cycles. To address this issue, future versions of the ONCA should incorporate an internal bundling mechanism that does not interfere with the needle's wasp-inspired self-propelled motion [11].

The design of the actuation system, particularly the needle segment advance mechanisms, constrains the needle segment diameter to 0.3 mm. This diameter constraint prohibits the integration of a needle or functional segment with a different diameter into the system. Consequently, a direct experimental comparison between the wasp-inspired segmented needle and a conventional needle using the ONCA is currently not possible. Nevertheless, a previous study has shown that a wasp-inspired self-propelled motion can reduce tissue motion and deformation as compared to needle insertion of a conventional needle [5]. In future *ex vivo* experiments using the ONCA, tissue damage can be evaluated through histological analysis, similar to the approach used by Gidde *et al.* [20,21], to enable comparison with conventional needle insertion.

### 4.3. Recommendations and future research

In the experiments, we evaluated the ONCA's $\eta_c(C)$, $\eta_p(C)$, and $\eta_m(C)$. Our results show that in air and 5-wt% gelatin, the mean $\eta_c(C)$ was near 100%, whereas in 10-wt% gelatin, the mean $\eta_c(C)$ was lower. $\eta_c(C)$ is a characteristic of the

pencil lead advance mechanism used and can be increased by increasing the clamping force of the advance mechanism. We performed force measurements using a force gauge to quantify the clamping and insertion forces exerted on the needle segments by the actuation system. During actuation, the measured peak insertion force applied by the actuation system to the needle ranged from 0.5 N to 0.8 N (0.7 ± 0.1 N, mean ± standard deviation, number of repetitions = 6). The measured peak clamping force applied to the individual locked needle segments ranged from 0.7 N to 2.9 N (1.6 ± 0.8 N, mean ± standard deviation, number of repetitions = 6). Increasing the clamping force can help resist forces acting on the needle segment within the stiffer substrate. The clamping force of the advance mechanism can be increased, for instance, by increasing the spring's stiffness or by reducing the taper angle of the sleeve. Nevertheless, the clamping force should not have such a high value that the needle segment deforms. This risk of deformation or damage increases when the needle segment is replaced with a functional segment, such as a hollow tube or optical fiber.

In order to enhance $\eta_p(C)$ of the ONCA, the friction between the stationary needle segments and the surrounding substrate can be increased by incorporating a directional friction surface topography on the needle shaft, as shown by Frasson *et al.* [22], Parittotokkaporn *et al.* [23], and Fung-A-Jou *et al.* [24]. Another method to increase $\eta_p(C)$ involves decreasing the friction and cutting forces on the advancing needle segment by, for example, sharpening the needle segment tips to lancet points [25,26] or inserting the needle segment via a rotational or a vibratory insertion motion [27,28]. Additionally, $\eta_p(C)$ can be increased by preventing the separation of the needle segments by incorporating an internal bundling mechanism [11].

To clinically use the ONCA as a passageway to a target location deep within the body, a functional segment can replace the central needle segment, as illustrated in Fig 9a. The functional segment can be an optical fiber for prostate cancer focal laser ablation. Focal laser ablation is a prostate cancer treatment option that leads to homogeneous tissue necrosis caused by an optical fiber that is positioned near the tumor using a needle [29]. Fig 9b–d demonstrate the ability of the needle to successfully propel out of various fruits with differing stiffnesses and inhomogeneous anatomies, with the optical fiber serving as the central needle segment. In all cases, the needle was initially inserted halfway through the fruit and then actuated to propel itself further and out of the fruit. The self-propelled actuation enabled the needle to effectively penetrate all the fruits, including their relatively tough skins.

Besides functionalization, addressing sterilization and image guidance techniques is essential. The ONCA is a modular design that can be a so-called "reposable" device [30,31], which integrates disposable and reusable modules. The needle segments can be easily removed from the actuation system when the advance mechanisms are in their released state. In our design, the needle that is in contact with the patient can be disposable, while the actuation system can be reusable. The main challenge that remains is the interconnection between the reusable actuation system and disposable needle segments. Research into a sterile barrier around the actuation system will be incorporated into future prototypes of our design. Furthermore, image guidance techniques can be used to visualize the trajectory of the needle inside the tissue, and implementing a feedback control system within the actuation system can enable real-time correction of this trajectory.

Our experiments were conducted in a controlled environment using tissue-mimicking phantoms made with gelatin powder. However, to assess the ONCA's functioning in a clinical setting, *ex vivo* or *in vivo* experiments are needed. These experiments will help us to investigate the effects of inhomogeneous tissue properties and the presence of different tissue layers (e.g., skin, fat, and muscle) with other mechanical properties (e.g., modulus of elasticity), as well as blood, on the self-propelling performance of the needle. During our evaluation, the needle was manually inserted 30 mm into the substrate before being actuated. In clinical practice, the needle must first puncture the skin of the patient before being able to self-propel through the tissue. The skin introduces a surface stiffness force due to the needle puncturing the skin until the moment of puncture [1]. To overcome this, manual insertion of the needle through the skin could be an option using an initial puncture needle, ensuring sufficient contact between the needle segments and the surrounding tissue for the self-propelled motion. The ability of a wasp-inspired self-propelled needle to advance in *ex vivo* porcine liver and human prostate tissue has been exemplified successfully in previous studies [9,10]. Moreover, Scali *et al.* [7] showed that a

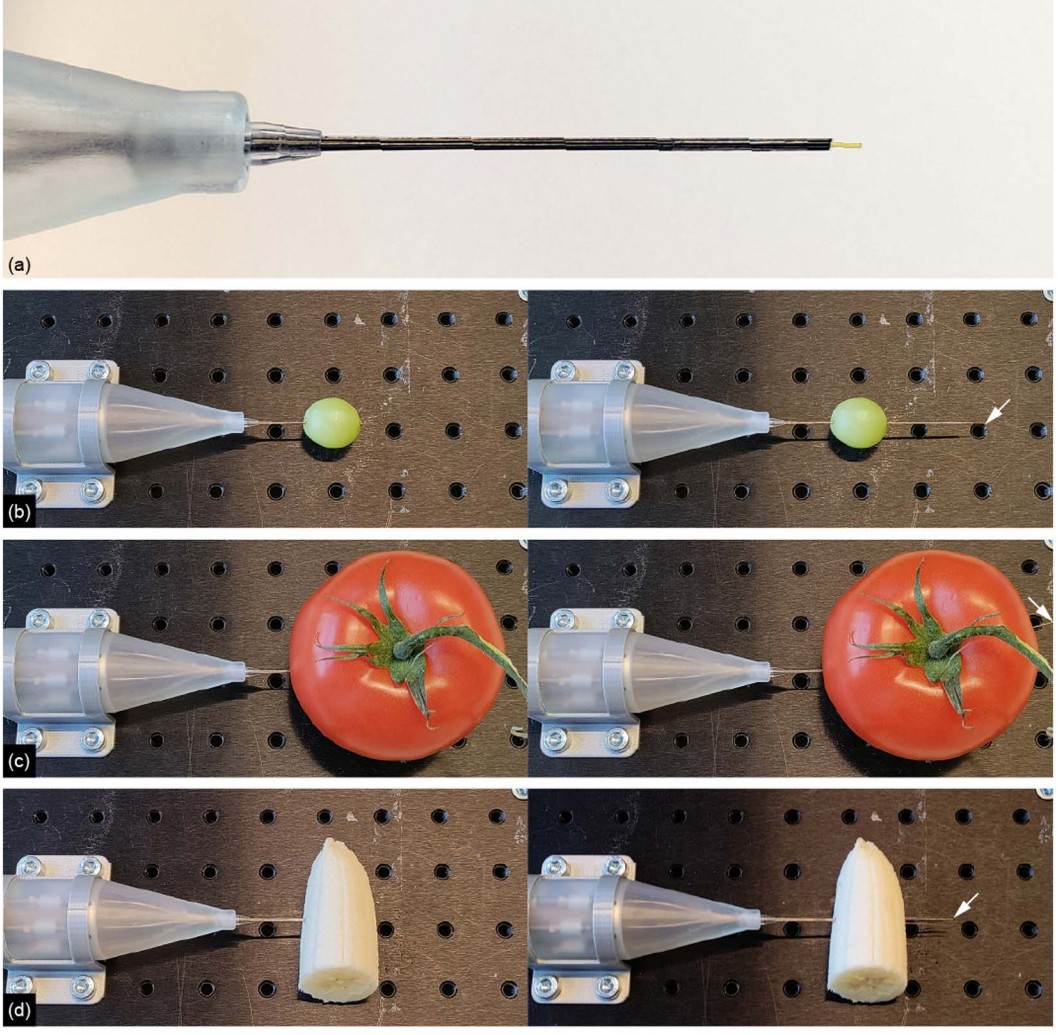

**Fig 9. Demonstration of the needle with optical fiber as central needle segment propelling through fruits with varying stiffnesses. (a)** Close-up of the needle tip consisting of six stainless steel rods and a central optical fiber as the seventh central needle segment. The pictures on the left show the initial position of the needle halfway inside the fruit before actuation. The pictures on the right show the needle propelled through the fruit after actuation, where the arrow marks the needle tip. Fruits: **(b)** grape, **(c)** tomato, and **(d)** banana.

wasp-inspired self-propelled needle can advance in multilayered tissue-mimicking phantoms. These previous studies suggest that the ONCA can achieve a similar performance. To conclusively demonstrate this, evaluating the ONCA in *ex vivo* and *in vivo* experiments is necessary.

## 5. Conclusion

This study presents the design and evaluation of a stationary actuation system that can advance a needle through stationary tissue-mimicking phantoms. Based on the pencil lead advance mechanism in a mechanical pencil that advances the pencil lead at a fixed increment when the pencil button is pressed, our actuation system advances the seven needle segments that comprise our needle by locking, advancing, releasing, and retracting the advance mechanisms. This actuation system allows advancing a wasp-inspired self-propelled needle with a theoretically unlimited insertion length.

Experimental evaluation revealed that the needle can self-propel in air, 5-wt% gelatin, and 10-wt% gelatin. The prototype's mean clamp efficiency in 5-wt% gelatin remained approximately constant and comparable to that of the needle in the air over the 20 actuation cycles. Meanwhile, in 10-wt% gelatin, the measured clamp efficiency decreased with increasing number of actuation cycles. In conclusion, the mechanical pencil-based actuation system is a step forward in developing self-propelled needles for targeting deep tissue structures.

## Nomenclature

| Symbol | Description | Unit |
|---|---|---|
| $a$ | Number of advancing needle segments | dimensionless |
| $C$ | Number of actuation cycles | dimensionless |
| $d_c$ | Traveled distance of the cart with respect to the air track | mm |
| $d_e$ | Expected traveled distance of the needle tip in the substrate | mm |
| $d_n$ | Traveled distance of the needle tip in the substrate | mm |
| $F_{cut,adv}$ | Cutting force on the tip of the advancing needle segments | N |
| $F_{fric,adv}$ | Friction force along the shafts of the advancing needle segments | N |
| $F_{fric,non-adv}$ | Friction force along the shafts of the non-advancing needle segments | N |
| $K$ | Spring constant | N/mm |
| $L_c$ | Spring length at maximum compression | mm |
| $L_0$ | Free spring length | mm |
| $n$ | Number of non-advancing needle segments | dimensionless |
| $s_{ratio}$ | Slip ratio of the slip between the stationary needle segments and the substrate | dimensionless |
| $x_n$ | Needle insertion distance | mm |
| $\varnothing_{outer}$ | Outer diameter | mm |
| $\varnothing_{wire}$ | Wire diameter | mm |
| $\eta_c$ | Clamp efficiency of the advance mechanisms inside the actuation system of the ONCA | % |
| $\eta_m$ | Motion efficiency of the ONCA | % |
| $\eta_p$ | Propulsion efficiency of the needle of the ONCA | % |

## Supporting information

**S1 Data. Raw data set of the experiments.**
(XLSX)

## Author contributions

**Conceptualization:** Jette Bloemberg, Aimée Sakes, Paul Breedveld.

**Data curation:** Jette Bloemberg.

**Formal analysis:** Jette Bloemberg.

**Funding acquisition:** Paul Breedveld.

**Investigation:** Jette Bloemberg.

**Methodology:** Jette Bloemberg, Aimée Sakes, Paul Breedveld.

**Project administration:** Paul Breedveld.

**Resources:** Jette Bloemberg, Mario van der Wel.

**Supervision:** Aimée Sakes, Paul Breedveld.

**Visualization:** Jette Bloemberg.

**Writing – original draft:** Jette Bloemberg.

**Writing – review & editing:** Mario van der Wel, Aimée Sakes, Paul Breedveld.

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
