## [Decision Letter · Decision Letter 0]

Dear Dr. Bloemberg,

Thank you for submitting your manuscript to PLOS ONE. After careful consideration, we feel that it has merit but does not fully meet PLOS ONE’s publication criteria as it currently stands. Therefore, we invite you to submit a revised version of the manuscript that addresses the points raised during the review process.

We look forward to receiving your revised manuscript.

Kind regards,

Bonnie Gray

Academic Editor

PLOS ONE

Journal Requirements:

Reviewers' comments:

Reviewer's Responses to Questions

**Comments to the Author**

1. Is the manuscript technically sound, and do the data support the conclusions?

Reviewer #1: Yes

Reviewer #2: Partly

2. Has the statistical analysis been performed appropriately and rigorously?

Reviewer #1: Yes

Reviewer #2: No

3. Have the authors made all data underlying the findings in their manuscript fully available?

Reviewer #1: Yes

Reviewer #2: Yes

4. Is the manuscript presented in an intelligible fashion and written in standard English?

Reviewer #1: Yes

Reviewer #2: Yes

Reviewer #1: This paper presents series of design extensions to the existing biologically inspired needle insertion mechanisms. The results of the paper are well-demonstrated combines with several invitro experimental studies using prototype mechanisms. The following are some comments which can be followed in order to better clarifies or expand on some of the stated claims of the paper.

• The paper uses 7 needle segments instead of 6 which was previous proposed in the literature. However, more theoretical motivation for such increased in the number of needles should be added. This can be in terms of reduced the effect of buckling of the needle or in reducing the overall friction force effects.

• There are many inter-related definitions of various terms used in the paper. It is to the paper’s advantage if a list of nomenclature be stated in the paper. For example, the term motion efficiency is used in the paper which one can assume related to the product of the other two efficiency state in section 3.3. However, motivation and significant for such efficiency in relation to standard conventional approaches should be stated clearly. Other terms such as central functional and functional segment. The list helps the reader for more accurate referencing the terms.

• How does actuation cycles be related to efficiency? With there be a calibration cycle or mechanical tuning of the system after certain cycle of usage?

• A mechanical design comparison table would be useful in relation to references 4-13 and reference [7] in relation to lower number of needles being used in comparison with the proposed method of seven needles.

• The clamping action with the sequence of cam rotation is not clear given the Figure 2 and 4. In addition, it would be helpful to show the 3D profile of the cam on how the groove where the follower can trace will look like. This way, it becomes clear on how the track on then cam is used by the slides in Figure 4 during the lock, advance, release and retract sequences.

• The notion of S-shaped tudes (seven of them) and how it connect/guide to the needle as shown in Figure 5-6 needs to be properly exploited.

Reviewer #2: This is an interesting paper, combining a bio-mimicking mechanism inspired by the wasp with the mechanism from a mechanical propelling pencil. However, there remain several questions that need addressing, detailed below:

1. There is a lack of simulation data regarding the forces experienced by the needle when inserting into tissue. A stated aim of the work is to overcome buckling issues. Simulations would provide relevant information on the multi segmented needles’ ability to withstand lateral and back forces during insertion.

There are many models in the literature, especially for micro needles inserting into tissue, that would provide more clarity on the issues faced by this system.

2. The experimental section does not provide any force data, merely travelled distance. Some form of mechanical testing using a universal Test Machine (UTM) would provide dots on this. Also, the applied force of the propelling machismo isn’t quantified.

3. The experimental section provides no comparisons to current needles. The segmented wasp inspired needle could be replaced in the propelling mechanism the authors have created with an equivalent diameter stainless steel needle (e.g. 18 gauge), and the same insertion experiments performed. This would clearly demonstrate the differences between the two needle types, showcasing the deflection of a normal needle compared to the wasp needle.

4. While there is some discussion around this in the paper, it is not clear how effective this method is for its intended use. If it’s for fluid extraction, will the repeated external pressing cause compression damage to the hollow needle at the centre, reducing or blocking flow? It it’s for biopsy, can sufficient force be applied to obtain the relevant biopsy sample without overcoming the friction lock of the propelling mechanism? Some force analysis should be provided, and the discussion expended.

5. There is a lack of repeated experiments. Fig 8 shows error bars related to mean values, but it appears to relate to the number of actuation cycles in a single insertion test. How many repeat insertion experiments were done with the same needle? As the needle is complicated to make, repeat testing would show how resistant the sliding mechanism of the needle is to fluids/tissue contamination, which could clog it.

6. Related to pint 5, as the needle is complicated to make, is it reusable? Can it be cleaned, sterilized after use? Or is it envisioned to be single use? If so, how does the cost to make it compare to using a standard needle? How easy is it to put on a new needle into the system?

7. In the experimental section, why was the process stopped after 60s? Does this indicate the needle sections did not extend fully as they met resistance? Could a feedback mechanism in the mechanical actuation section be included to detect such resistance?

8. Line 302 - why was insertion done manually at first?

9. The discussion mentions a needle of infinite length can be used with the system, but that will require multiple needle segments to effect the wasp bio-mimicking property, which will increase the overall needle diameter and add complexity to the driving mechanism, ultimately limiting it’s practicality. The authors should clarify their comment.

10. In line 363, forces are mentioned as a limiting factor, but it’s not discussed if the tip was sharp, or blunt, which will affect the frictional forces. The description was of a sever stainless steel rods, implying the tip was blunt. The authors do state in their future work the sharp tips should be used, but this needs to be clearer here. Also, a method to overcome this would be to increasese the clamping force. This is discussed in lines 411 to 418, but as it was known from the literature already, why were such measures not incorporated? Additionally, would increased clamping force cause damage to a hollow needle if used in this process?

11. On line 423, the phrase ‘The self-propelled actuation enabled the needle to effectively penetrate all the fruits, including their relatively tough skins’ is confusing as it’s also stated that the needle was ‘initially inserted halfway through the fruit and then actuated to propel itself further’. Does this mean the needle self propelled itself out of the fruit’s skin, rather than into it? Exiting is easier than entering, as the needle is supported by the internal material and experiences less lateral buckling forces. Also, the phrase ‘tough’ skin should be quantified. Skin on a tomato is taut, which makes it easier to penetrate than human skin, which is

highly elastic. More commentary on the experimental processes used should be made, including why a standard human tissue analog, pig skin, was not used.

12. Lastly, there is a lot of unnecessary duplication of the results in multiple sections (discussion, limitations and conclusions).

**Do you want your identity to be public for this peer review?** For information about this choice, including consent withdrawal, please see our Privacy Policy

Reviewer #1: **Yes: ** Shahram Payandeh

Reviewer #2: No

---

## [Author Response · Author response to Decision Letter 1]

28 Apr 2025

Dear Bonnie Gray,

Thank you for your time and consideration of our manuscript. Based on your editorial comments and the reviewers’ comments below, we have revised the manuscript accordingly.

Regarding the points described in your email, we confirm that (1) we adapted the manuscript to meet PLOS ONE’s style requirements, and (2) we reviewed the reference list to ensure it is complete and correct. Reference [11] was removed from the initial manuscript, as it was the same as reference [13]. References [20, 21, 29-31] were added in the revised manuscript as part of revisions R2.3, R2.4, and R2.6.

11. Bloemberg J, van Wees S, Kortman VG, Sakes A. Design of a Wasp-Inspired Biopsy Needle Capable of Self-Propulsion and Friction-Based Tissue Transport. Frontiers in Bioengineering and Biotechnology.12:1497221. doi: 10.3389/fbioe.2024.1497221

13. Bloemberg J, van Wees S, Kortman VG, Sakes A. Design of a wasp-inspired biopsy needle capable of self-propulsion and friction-based tissue transport. Frontiers in Bioengineering and Biotechnology. 2025;12:1497221. doi: 10.3389/fbioe.2024.1497221.

20. Gidde STR, Acharya SR, Kandel S, Pleshko N, Hutapea P. Assessment of tissue damage from mosquito-inspired surgical needle. Minimally Invasive Therapy & Allied Technologies. 2022;31(7):1112-21. doi: 10.1080/13645706.2022.2051718.

21. Gidde STR, Islam S, Kim A, Hutapea P. Experimental study of mosquito-inspired needle to minimize insertion force and tissue deformation. Proceedings of the Institution of Mechanical Engineers, Part H: Journal of Engineering in Medicine. 2023;237(1):113-23. doi: 10.1177/09544119221137133.

29. Oto A, Sethi I, Karczmar G, McNichols R, Ivancevic MK, Stadler WM, et al. MR imaging–guided focal laser ablation for prostate cancer: phase I trial. Radiology. 2013;267(3):932-40. doi: 10.1148/radiol.13121652.

30. Malchesky PS, Chamberlain VC, Scott-Conner C, Salis B, Wallace C. Reprocessing of reusable medical devices. ASAIO journal. 1995;41(2):146-51. doi: n/a.

31. Abreu EL, Haire DM, Malchesky PS, Wolf-Bloom DF, Cornhill JF. Development of a program model to evaluate the potential for reuse of single-use medical devices: results of a pilot test study. Biomedical instrumentation & technology. 2002;36(6):389-404. doi: 10.2345/0899-8205(2002)36[389%3ADOAPMT]2.0.CO%3B2.

Please find our detailed responses to the reviewers below. In the revised manuscript, the changes are indicated using track changes.

Review Comments to the Author

Reviewer #1: This paper presents series of design extensions to the existing biologically inspired needle insertion mechanisms. The results of the paper are well-demonstrated combines with several invitro experimental studies using prototype mechanisms. The following are some comments which can be followed in order to better clarifies or expand on some of the stated claims of the paper.

• The paper uses 7 needle segments instead of 6 which was previous proposed in the literature. However, more theoretical motivation for such increased in the number of needles should be added. This can be in terms of reduced the effect of buckling of the needle or in reducing the overall friction force effects.

R1.1. Thank you for your compliments on our work and your suggestion to add a theoretical motivation for the use of seven needle segments instead of six. In the revised Section 2.1, in Lines 84-89 we added an explanation of why we used seven needle segments in our needle design. In the two-dimensional cross-section, concentrically arranging six cylindrical needle segments around the seventh needle segment forms an optimal configuration when all needle segments have the same diameter. Each outer needle segment contacts the central needle segment as well as two adjacent needle segments. This minimizes the total cross-sectional area and results in the least empty space between the needle segments where tissue could accumulate.

Please also refer to R1.4. for a comparison of the number of needle segments for the state-of-the-art in wasp-inspired needles.

• There are many inter-related definitions of various terms used in the paper. It is to the paper’s advantage if a list of nomenclature be stated in the paper. For example, the term motion efficiency is used in the paper which one can assume related to the product of the other two efficiency state in section 3.3. However, motivation and significant for such efficiency in relation to standard conventional approaches should be stated clearly. Other terms such as central functional and functional segment. The list helps the reader for more accurate referencing the terms.

R1.2. Thank you for your useful suggestion on adding a list of nomenclature. We have added a list of nomenclature at the start of the manuscript in Line 28. Moreover, we replaced “advanced position” in Lines 102-103, 111, 185-186 with “propelled position” to be consistent, and we replaced “functional element” in Line 461 in Section 4.3 with “functional segment”.

• How does actuation cycles be related to efficiency? With there be a calibration cycle or mechanical tuning of the system after certain cycle of usage?

R1.3. In this study, we define one actuation cycle as the process where all seven needle segments move forward once. This happens when the cam inside our prototype completes one full rotation. Each full rotation of the cam is supposed to push the needle segments forward by a certain amount, i.e., the needle stroke distance. That is the distance the needle segments are expected to travel into the substrate. In our experiments, we wanted to see how well our prototype performed when it moved through the substrate, specifically, how efficient the motion is after a certain number of actuation cycles. Therefore, we looked at whether the needle segments slipped (slid unintentionally) when they interacted with:

the advance mechanisms that push them forward,

the substrate into which they are moving.

Slipping reduces how far the needle segments advance into the substrate as compared to what we expect. To clarify the definition of actuation cycle and its relation to the needle stroke distance, we added in Section 2.3, Lines 202-204, that one actuation cycle is defined as a single complete rotation of the cam, during which all seven needle segments are advanced once over the needle stroke distance.

• A mechanical design comparison table would be useful in relation to references 4-13 and reference [7] in relation to lower number of needles being used in comparison with the proposed method of seven needles.

R1.4. We want to thank the reviewer for this suggestion. We, therefore, added Table 1 in Lines 67-69, which shows a comparison of the state-of-the-art in wasp-inspired needles, showing the number of needle segments the complete needle comprises, the outer diameter of the needle, the material of the needle segments, and a description of the needle design.

• The clamping action with the sequence of cam rotation is not clear given the Figure 2 and 4. In addition, it would be helpful to show the 3D profile of the cam on how the groove where the follower can trace will look like. This way, it becomes clear on how the track on then cam is used by the slides in Figure 4 during the lock, advance, release and retract sequences.

R1.5. In Figure 5, we added subfigure 5d, which shows the 3D cam with one of the seven advance mechanisms in the cam tracks and the 3D rolled-out cam. The colors of the cam in Fig 5d are consistent with the colors in Figs 2 and 4 to clearly indicate the lock, advance, release, and retract phases.

• The notion of S-shaped tudes (seven of them) and how it connect/guide to the needle as shown in Figure 5-6 needs to be properly exploited.

R1.6. In the revised Section 2.3 in Lines 212-214, we added that to guide the seven needle segments smoothly from the actuation system to the needle tip, seven S-shaped tubes followed by seven straight tubes were used. Furthermore, we added in Lines 214-216 that the needle segments run through the S-shaped tubes.

Reviewer #2: This is an interesting paper, combining a bio-mimicking mechanism inspired by the wasp with the mechanism from a mechanical propelling pencil. However, there remain several questions that need addressing, detailed below:

1. There is a lack of simulation data regarding the forces experienced by the needle when inserting into tissue. A stated aim of the work is to overcome buckling issues. Simulations would provide relevant information on the multi segmented needles’ ability to withstand lateral and back forces during insertion.

There are many models in the literature, especially for micro needles inserting into tissue, that would provide more clarity on the issues faced by this system.

R2.1. Thank you for suggesting modeling the forces when inserting a needle into the tissue to provide more clarity on the issues faced by the needle system. In the revised Section 1, in Lines 34-36, we added that Okamura et al. (2004) demonstrated that inside homogeneous tissue, the forces at the needle tip and along the needle shaft comprise cutting and friction forces. To move a needle through tissue, the operator should apply a force that overcomes the sum of these forces acting on the needle. Moreover, in Lines 52-57, we added Eq 1, which must hold to achieve the wasp-inspired self-propelled motion.

∑_(i=1)^a▒(F_(fric,adv,i)+F_(cut,adv,i) ) ≤∑_(j=1)^n▒(F_(fric,non-adv,j) ) (1)

Where a is the number of advancing needle segments, n is the number of non-advancing needle segments, F_(fric,adv) is the friction force along the advancing needle segments, F_(cut,adv) is the cutting force on the tip of the advancing needle segments, and F_(fric,non-adv) is the total amount of friction of the non-advancing needle segments, which works in the opposite direction to the friction force of the advancing needle segments.

Okamura AM, Simone C, O'leary MD. Force modeling for needle insertion into so� tissue. IEEE transactions on biomedical engineering. 2004;51(10):1707-16. doi: 10.1109/TBME.2004.831542.

2. The experimental section does not provide any force data, merely travelled distance. Some form of mechanical testing using a universal Test Machine (UTM) would provide dots on this. Also, the applied force of the propelling machismo isn’t quantified.

R2.2. Thank you for your suggestion to provide information on the force applied by the actuation system. In our design, the force applied by the actuation system is caused by the springs in the needle segment advance mechanisms. Therefore, in the revised Section 2.4, in Lines 237-238, we added the spring characteristics of the springs used in the needle segment advance mechanisms: DR970 springs (Alcomex springs, Opmeer, Netherlands; ∅_outer=3.6 mm, ∅_wire=0.4 mm, L_0=12.80 mm, K=0.78 N/mm, L_c=5.50 mm). Moreover, we performed force measurements using a force gauge to quantify the clamping and insertion forces exerted on the needle segments by the actuation system. We added the results of these force measurements in the revised Section 4.3, in Lines 441-446. During actuation, the measured peak force applied by the actuation system to the needle bundle ranged from 0.5 N to 0.8 N (0.7 ± 0.1 N, mean ± standard deviation, n = 6). The measured clamping force applied to the individual locked needle segments ranged from 0.7 N to 2.9 N (1.6 ± 0.8 N, mean ± standard deviation, n = 6),

3. The experimental section provides no comparisons to current needles. The segmented wasp inspired needle could be replaced in the propelling mechanism the authors have created with an equivalent diameter stainless steel needle (e.g. 18 gauge), and the same insertion experiments performed. This would clearly demonstrate the differences between the two needle types, showcasing the deflection of a normal needle compared to the wasp needle.

R2.3. Thank you for your suggestion. A direct comparison between our segmented wasp-inspired needle and a conventional stainless-steel needle would help to highlight the advantages of our design. However, due to the design of our actuation system, the needle segment diameter that fits in the needle segment advance mechanisms is limited to 0.3 mm. This constraint makes it infeasible to replace the segmented needle with a single conventional needle of equivalent overall diameter (~1.0 mm). The actuation mechanism was designed to accommodate up to seven 0.3-mm diameter needle segments and cannot accommodate a larger-diameter needle.

Nevertheless, to address this point, we have added a paragraph in the discussion in Section 4.2, in Lines 428-436, explaining this constraint, along with a comparison based on the needle behavior reported in the scientific literature. Our needle uses a wasp-inspired sequential actuation mechanism that may reduce insertion force and tissue deformation compared to conventional needles that are pushed through the tissue. In future ex vivo experiments using the ONCA, tissue damage can be evaluated through histological analysis, similar to the approach used by Gidde et al. (2022, 2023), to enable comparison with conventional needle insertion.

Gidde STR, Acharya SR, Kandel S, Pleshko N, Hutapea P. Assessment of tissue damage from mosquito-inspired surgical needle. Minimally Invasive �erapy & Allied Technologies. 2022;31(7):1112-21. doi: 10.1080/13645706.2022.2051718.

Gidde STR, Islam S, Kim A, Hutapea P. Experimental study of mosquito-inspired needle to minimize insertion force and tissue deformation. Proceedings of the Institution of Mechanical Engineers, Part H: Journal of Engineering in Medicine. 2023;237(1):113-23. doi: 10.1177/09544119221137133.

4. While there is some discussion around this in the paper, it is not clear how effective this method is for its intended use. If it’s for fluid extraction, will the repeated external pressing cause compression damage to the hollow needle at the centre, reducing or blocking flow? It it’s for biopsy, can sufficient force be applied to obtain the relevant biopsy sample without overcoming the friction lock of the propelling mechanism? Some force analysis should be provided, and the discussion expended.

R2.4. Thank you for pointing out this lack of clarity. The intended use of our needle is for positioning functional segments, such as optical fiber, for prostate cancer focal laser ablation. We elaborated on this in the revised discussion in Section 4.3, in Lines 461-464. Focal laser ablation is a prostate cancer treatment option that leads to homogeneous tissue necrosis caused by an optical fiber that is positioned near the tumor using a needle (Oto et al., 2013). Therefore, we demonstrated in the additional test in fruits the ability of the needle to successfully self-propel with the optical fiber serving as the central needle segment.

Oto A, Sethi I, Karczmar G, McNichols R, Ivancevic MK, Stadler WM, et al. MR imaging–guided focal laser ablation for prostate cancer: phase I trial. Radiology. 2013;267(3):932–40. pmid:23440319

5. There is a lack of repeated experiments. Fig 8 shows error bars related to mean values, but it appears to relate to the number of actuation cycles in a single insertion test. How many repeat insertion experiments were done with the same needle? As the needle is complicated to make, repeat testing would show how resistant the sliding mechanism of the needle is to fluids/tissue contamination, which could clog it.

R2.5. Thank you for pointing out this missing information in the manuscript. We repeated each experimental condition six times in order to compute the mean values and the standard deviations (over six insertion tests) reported in Table 2. We added this information in Section 3.4, in Lines 328-329.

6. Related to pint 5, as the needle is complicated to make, is it reusable? Can it be cleaned, sterilized after use? Or is it envisioned to be single use? If so, how does the cost to make it compare to using a standard needle? How easy is it to put on a new needle into the system?

R2.6. Thank you for your questions. In the revised Section 4.3, in Lines 476-485, we added a paragraph on sterilizing our design. The ONCA is a modular design that can be a so-called “reposable” device (Malchesky et al., 1995; Abreu et al., 2002), integrating disposable and reusable modules. The needl

---

## [Decision Letter · Decision Letter 1]

Design and evaluation of a mechanical pencil-based actuator for a wasp-inspired needle

PONE-D-25-11416R1

Dear Dr. Bloemberg,

We’re pleased to inform you that your manuscript has been judged scientifically suitable for publication and will be formally accepted for publication once it meets all outstanding technical requirements.

Kind regards,

Bonnie Gray

Academic Editor

PLOS ONE

Additional Editor Comments (optional):

Reviewers' comments:

Reviewer's Responses to Questions

**Comments to the Author**

Reviewer #1: All comments have been addressed

Reviewer #2: All comments have been addressed

2. Is the manuscript technically sound, and do the data support the conclusions?

Reviewer #1: Yes

Reviewer #2: Yes

3. Has the statistical analysis been performed appropriately and rigorously?

Reviewer #1: N/A

Reviewer #2: Yes

4. Have the authors made all data underlying the findings in their manuscript fully available?

Reviewer #1: Yes

Reviewer #2: Yes

5. Is the manuscript presented in an intelligible fashion and written in standard English?

Reviewer #1: Yes

Reviewer #2: Yes

Reviewer #1: authors have addressed suggested revisions in particular regarding the actuation mechanism and the buckling conditions.

Reviewer #2: (No Response)

**Do you want your identity to be public for this peer review?** For information about this choice, including consent withdrawal, please see our Privacy Policy

Reviewer #1: **Yes: ** Shahram Payandeh

Reviewer #2: No

---

## [Editor Report · Acceptance letter]

PONE-D-25-11416R1

PLOS ONE

Dear Dr. Bloemberg,

I'm pleased to inform you that your manuscript has been deemed suitable for publication in PLOS ONE. Congratulations! Your manuscript is now being handed over to our production team.

Kind regards,

on behalf of

Dr. Bonnie Gray

Academic Editor

PLOS ONE